# COVID-19 Vaccine and Long COVID: A Scoping Review

**DOI:** 10.3390/life12071066

**Published:** 2022-07-16

**Authors:** Aqsa Mumtaz, Abdul Ahad Ehsan Sheikh, Amin Moazzam Khan, Subaina Naeem Khalid, Jehanzaeb Khan, Adeel Nasrullah, Shazib Sagheer, Abu Baker Sheikh

**Affiliations:** 1Department of Internal Medicine, Montefiore St. Luke’s Cornwall Hospital, Newburgh, NY 12550, USA; aqsamumtazmd@gmail.com; 2Department of Internal Medicine, The Wright Center for Graduate Medical Education, Scranton, PA 18505, USA; abdulahad.esn@gmail.com; 3Department of Internal Medicine, Shifa College of Medicine, Islamabad 46000, Pakistan; aminmoazzamkhan@gmail.com (A.M.K.); subainakhalid@gmail.com (S.N.K.); 4Department of Internal Medicine, University of New Mexico Health Sciences Center, Albuquerque, NM 87102, USA; jjkhan@salud.unm.edu; 5Division of Pulmonary and Critical Care, Allegheny Health Network, Pittsburgh, PA 15222, USA; adeel.shifa@gmail.com; 6Division of Cardiology, University of New Mexico Health Sciences Center, Albuquerque, NM 87102, USA; ssagheer@salud.unm.edu

**Keywords:** long COVID, COVID-19 vaccination, long COVID treatment, COVID-19

## Abstract

As the COVID-19 pandemic progresses, changing definitions and therapeutics regarding the post-acute sequela of COVID-19, particularly long COVID, have become a subject of great interest and study. The study aims to describe the pathophysiology and discuss different therapeutic agents currently available for long COVID. Another objective is to assess comparative efficacy between different types of vaccines on symptoms of long COVID. A preliminary search was conducted using Ovid Medline, Embase, medRxiv, and NIH COVID-19 portfolios. A total of 16 studies were included in our review. Despite some of the data showing variable results, most of the vaccinated patients reported improvement in long COVID symptoms with no significant difference between various types of vaccines. Further trials are needed to better identify the comparative efficacy of vaccines for long COVID and ascertain other therapeutic modalities.

## 1. Introduction

As the world continues to struggle against the coronavirus disease of 2019 (COVID-19), long-term effects of the disease have started to emerge. Wisely predicted as a pandemic after a pandemic, long COVID has been studied both as post-acute sequelae (PAS) and as a post-COVID-19 condition (PCC). The World Health Organization has defined long COVID as “A condition which occurs in individuals with a history of probable or confirmed severe acute respiratory syndrome coronavirus 2 (SARS-CoV-2) infection, usually three months from the onset of COVID-19 with symptoms that last for at least two months and cannot be explained by an alternative diagnosis” [1]. Long COVID can impact any organ in the body and is characterized by a wide range of symptoms, profound fatigue being the most common one [2]. Uncertainties around the etiology and pathophysiology of long COVID have raised alarm in the general population and have led to vaccine hesitancy in the general population [3]. While vaccinations have significantly proven to be efficacious in decreasing the severity of symptoms and overall hospitalization rate, our understanding of their effect on long COVID remains limited [4,5].

The main objectives of this study are:To describe the pathophysiology of long COVID along with the most common signs and symptoms.To study the impact of vaccination on patients with preexisting signs and symptoms of long COVID.To decipher the impact of vaccinations before developing COVID-19 infection.To find any possible differences that may exist in the rates of developing long COVID when comparing vaccinated vs. unvaccinated people.To distinguish any similarities or differences between the various types of vaccines and the risk of developing long COVID symptoms.To report the management and therapeutic options available for long COVID.

## 2. Methods

A preliminary search was carried out using Ovid Medline, Embase, and NIH COVID-19 portfolios as well as directly searching medRxiv by three reviewers. Subsequently, title and abstract screening, as well as full-text screening with the interpretation of study results, were performed using specific inclusion and exclusion criteria. Articles published between 1 January 2020 to 20 April 2022 were included. Only those articles that were either originally in English or translated to English were included. Search terms used reflected key features of the review question. These included “vaccine”, “vaccination, efficacy”, “post COVID syndrome”, “long COVID”, “post-acute sequelae of COVID-19”, “Moderna”, “Pfizer”, “AstraZeneca”, “J&J”, “Janssen”. Randomized controlled trials, Cohort studies, and case-control studies were included. Lastly, both peer-reviewed published papers as well as non-peer-reviewed preprints were incorporated into the review.

## 3. Results

### A Total of 16 Studies Were Included in Our Review and Were Divided into Two Groups

In the first group, 12 studies were evaluated regarding the effects of vaccinations on pre-existing COVID symptoms, as shown in Table 1. A total of 35,527 participants were included, with 32,726 of those vaccinated against SARS-CoV-2. A total of 8667 participants had preexisting signs and symptoms of long COVID. While there were limited data available about first and second doses, most of the participants reported improvement in symptoms after one dose, with few claiming no change, and a small cohort described worsening symptoms.

A prospective cohort study by Ayoubkhani et al. [8] utilized a questionnaire regarding the patient experience of long COVID symptomatology and functional status. A total of 28,356 participants were included and followed for an 8-month period. The first and second vaccine dose were associated with an improvement of long COVID symptomatology as defined in the study (OR = 0.87) and (OR = 0.91), respectively.

A cross-sectional study by Nehme et al. [11] recruited 1596 symptomatic patients who were followed by means of a survey for the development of six cardinal symptoms of long COVID as defined by the study. A total of 825 participants were unvaccinated, 424 were partially vaccinated, and 347 were fully vaccinated. Participants who were partially or fully vaccinated had a decreased prevalence of the six cardinal post-SARS-CoV-2 symptoms (adjusted OR = 0.72) as well as a decreased prevalence of any one symptom (adjusted OR = 0.60).

Strain et al. [15] conducted an online survey where patients with long COVID were invited to complete a web-based questionnaire that included the range and severity of long COVID symptoms and which vaccine they had received. Out of the 812 participants included, 698 were partially vaccinated and 114 were fully vaccinated. Of the total participants, 57% reported an improvement in symptoms, 25% reported no change, while 19% of participants reported a worsening of symptoms.

A prospective cohort study conducted by Tran et al. [16] recruited 910 participants who either had confirmed or highly suspected COVID-19 infection and were followed for 120 days. A total of 455 patients were vaccinated and were matched with 455 unvaccinated patients. The participant-reported data were then analyzed to form an arbitrarily assigned ST score from 0 to 53 reflecting the number and severity of symptoms. It was found that long COVID symptoms were less severe in vaccinated compared with unvaccinated participants (mean difference in COVID ST score = −1.8) and it was also found that vaccinated participants were more likely to experience remission of symptoms (HR = 1.97).

A prospective cohort study by Arnold et al. [7] recruited 44 vaccinated and 22 unvaccinated patients who were exhibiting long COVID symptoms. Both sets of participants were then followed for a period of one month for persistent symptoms of long COVID-19. The study found that vaccinated patients were more likely to show symptomatic improvement (23.2% vs. 15.4%, *p* = 0.035).

A prospective cohort study by Wisnivesky et al. [17] recruited 453 participants with at least one long COVID symptom to complete a baseline interview regarding symptomatology, who were then followed for 6 months. Of those participants, 324 were vaccinated and 129 were unvaccinated. By the end of the 6-month period, the study showed no significant association between symptomatology as defined by the study and vaccination status (*p* > 0.05 for all comparisons).

Scherlinger et al. [13] evaluated 380 vaccinated patients exhibiting long COVID symptoms for at least 4 weeks. The study found that 47% of patients demonstrated no change in symptoms over the 4-week period, 31% of patients showed worsening symptoms, and lastly only 22% of patients felt an improvement in their symptoms, most commonly anosmia and brain fog.

A prospective cohort study conducted by Gaber et al. [9] evaluated 83 healthcare workers who were suffering from long COVID syndrome. The study utilized a questionnaire that evaluated any changes in long COVID symptomatology over the course of several weeks. It was found that 14 (21%) participants reported symptomatic improvement, 8 (12%) participants reported a worsening in symptoms, and 45 (67%) participants reported no change in their symptoms.

A prospective cohort study conducted by Peghin et al. [12] investigated the impact of vaccination on the development of long COVID symptoms. Participants were to be evaluated at baseline and then telephone-interviewed by the same trained nurses at 6 and 12 months using a standardized questionnaire. Out of the 479 participants recruited, 132 patients were fully vaccinated, while 347 patients were unvaccinated. After analysis, the study found that there was no statistically significant difference between the vaccinated and unvaccinated groups regarding any of the post-COVID syndrome symptoms as defined in the study (*p* > 0.05).

Senjam et al. [14] conducted a cross-sectional study aimed to examine a total of 773 participants with confirmed COVID-19 infection by asking them to complete a survey over a 6-week period. Of those participants, 407 were unvaccinated, 173 were partially vaccinated, and 193 were fully vaccinated. The study found that fully vaccinated individuals were less likely to have long COVID symptoms (OR = 0.55).

A cross-sectional study nested in a prospective cohort study by Kuodi et al. [10] recruited 951 participants with confirmed COVID-19 infection (RT-PCR) over 1 year. Participants were then followed up between 16 July and 18 November 2021 by means of an online survey. Upon analysis, the study found that the fully vaccinated were 36–73% less likely to report 7 of the 10 most commonly reported long COVID symptoms (*p* < 0.04).

A retrospective cohort study conducted by Arjun et al. [6] recruited 487 participants who tested positive for COVID-19. Participants were surveyed regarding vaccination status and long COVID symptomatology. A total of 287 participants were fully vaccinated with COVAXIN, 81 participants were partially vaccinated, and 119 participants were unvaccinated. The study found a positive correlation between vaccination status and the development of long COVID symptoms (adjusted OR = 2.32).

The second group focused on the effects of vaccination on long COVID in patients who were vaccinated before getting infected with COVID-19 as shown in Table 2. A total of 264,389 participants were studied, with 32,052 vaccinated participants.

A retrospective cohort study was conducted by Simon et al. [20] in 2021 aimed at studying the relationship between vaccination status before or after confirmed COVID-19 infection and the development of long COVID symptoms. A total of 240,648 cases were eventually recruited. Of these, 220,460 cases were unvaccinated by 12 weeks after their COVID-19 diagnosis, 17,796 cases received 1 dose of vaccine within 12 weeks of their diagnosis, and only 2392 cases received 1 dose of vaccine before their diagnosis. Participants were followed and surveyed at 12 and 20-week periods after diagnosis for the persistence of long COVID symptoms. Participants who were vaccinated before diagnosis, 4 to 8 weeks after diagnosis, and 8 to 12 weeks after diagnosis were less likely to have any symptoms of long COVID after the follow-up period (OR = 0.220) (OR = 0.535) (OR = 0.747) respectively.

A nested case-control study by Antonelli et al. [18] recruited 4,740 participants and followed them for one month for the persistence of long COVID symptoms. A total of 2370 unvaccinated patients were matched with the same number of participants who were fully vaccinated. It was found that participants of all age groups who were partially or fully vaccinated were less likely to be hospitalized (OR = 0.31). The study also found that fully vaccinated participants of all groups were about half as likely to have symptoms lasting ≥ 28 days than unvaccinated participants (OR = 0.51).

Taquet et al. [19] recruited 9479 vaccinated individuals who were matched with the same number of unvaccinated individuals with confirmed COVID-19 infection. The participants were followed and surveyed after a 6-month period. The study found that there was no association between vaccination and the composite long COVID-19 outcome (HR = 1).

In this prospective cross-sectional study by Blumberg et al. [21], 28 unvaccinated and 15 vaccinated participants were recruited. All the individuals underwent a symptom limited cardio-pulmonary exercise test (CPET) after acute COVID-19 infection to compare aerobic capacity and exercise performance. A significant difference between the vaccinated and unvaccinated group was identified with a lower peak oxygen-consumption percentage, reduction in the peak-exercise heart rate, and lower ventilation values noted in the vaccinated group.

## 4. Discussion

Long COVID, an umbrella term used to define the persistent symptoms of COVID-19, recently sparked immense interest in the scientific society as the long-term implications of COVID-19 started to emerge. Almost two-thirds of COVID-19 survivors have reported one or more residual symptoms even after 3 months; social media first named this phenomenon long COVID. Long haulers, chronic COVID syndrome, post-COVID syndrome, PCC, and PAS are some of the terms used for the recurring and relapsing symptoms of COVID-19.

Long COVID is a multiorgan debilitating illness for the majority of those who experience it. The exact pathophysiology of long COVID still remains a mystery and is postulated to be multifactorial. Few hypotheses include viral- or immune-mediated organ injury, neurological involvement, dysautonomia, physical deconditioning, and psychological burden [22,23]. Persistent viremia, viral shedding, variable organ injury, and healing time are the few most common hypotheses. While the pathophysiology remains uncertain, there are also no definite diagnostic criteria for long COVID. Involvement of all organ systems has been reported and long COVID is characterized by a wide range of symptoms ranging from fatigue to cognitive dysfunctions. If anything, it highlights the heterogeneity of symptoms, and the significant functional impact of prolonged illness on somatic, cognitive, and psychological health.

Although symptoms of Long COVID vary in almost everyone, Sudre et al. have identified two patterns of symptoms in patients suffering from Long COVID. They were (1) pain, fatigue, and upper respiratory symptoms; or (2) fever and GI symptoms, in addition to the aforementioned symptoms. They also reported that the presence of five or more symptoms during the acute phase of COVID-19 was a strong indicator of developing long COVID. The most predictive five symptoms were fatigue, headache, dyspnea, hoarse voice, and myalgia regardless of gender [24].

On the contrary, Nikki et al. reported increasing age and female gender as the biggest risk factors of developing long COVID [25]. In literature, the severity of disease at onset, hospital admission, obesity, pre-existing comorbidities (particularly asthma), and the need for oxygen therapy are highlighted as the main determinants of long-term symptoms [26,27].

While most studies focused on the trend of developing long COVID after hospitalization, Ziauddeen et al. in a cross-sectional study assessed the chances of developing signs and symptoms of long COVID in patients who had mild-to-moderate symptoms or were not tested during the first wave of COVID. She reported that 77.7% of patients continue to feel one or more symptoms of long COVID, with most of them utilizing one or more healthcare facilities. This is a concerning finding and can have potentially dire implications on the already overburdened healthcare system. As such patients may be not prioritized for follow-up care, they can present with serious complications which might have been prevented altogether [28]. At the population level, it is critical to quantify the burden of long COVID to assess its impact on the healthcare system and identify age, gender, and the demographic group as probable risk factors to ensure inclusive, equitable, and effective healthcare for everyone despite the growing healthcare disparities.

It is hypothesized that persistent viral particles can cause molecular mimicry like those in rheumatic disease and can also dysregulate the immune system, leading to PCS [29]. Vaccination can help reset the immune and inflammatory response and can also help eradicate the persistent viral particles. More studies are needed to identify the triggers and if consistent can help develop the potential therapeutic targets.

### 4.1. COVID-19 Vaccination after Developing COVID-19 Infection

As COVID-19 began to engulf the world, a revolutionary preparation and distribution of different kinds of vaccines started. Although initially skeptical due to the hasty mass production of vaccines, they proved to be effective against the disease with a significant decrease in the severity of symptoms and hospitalization rate.

Arnold et al. studied hospitalized patients who developed long COVID and observed the effects of vaccination on their pre-existing signs and symptoms [7]. He reported a decrease in symptom burden in those patients who were vaccinated with no worsening of symptoms or quality of life. His findings were consistent with Ayoub, who observed that the first dose of vaccine was significantly associated with a 13% decrease in the development of long COVID symptoms with sustained improvement after the second dose over the median follow-up for 67 days [8]. These findings were consistent with other studies by Strain and Tran [15,16].

Nehme et al. also reported that vaccination decreases the six cardinal symptoms of long COVID [11]. These symptoms were fatigue, difficulty concentrating or memory loss, loss of or change in smell and taste, shortness of breath, and headache.

While the preliminary data are encouraging, it must be noted that in the studies above, a small proportion of participants reported either worsening of symptoms or no change in the symptoms whatsoever. Gaber et al. interestingly concluded that while the risk of symptoms aggravating persists, mRNA vaccines are twice as likely to improve rather than worsen the symptoms of long COVID [9].

### 4.2. COVID-19 Vaccination before Developing COVID-19 Infection

Simon et al. reported a significant decrease in developing signs and symptoms of Long COVID in the participants who had no prior history of COVID 19 infection and were given a single dose of vaccination [20]. Al-Aly et al. also identified a similar finding in the population who received both doses and had no history of prior infection [30]. On the contrary, Taquet et al. did not find any difference between vaccinated and unvaccinated populations [19]. Paradoxically, Arjun et al. reported worsening symptoms in participants who were completely vaccinated before contracting the coronavirus disease [6].

### 4.3. Difference between Vaccines?

The development of vaccines at a breakneck speed has led to questions around not only their efficacy and safety profile, but has also sparked a debate on which vaccine works best. Many papers have been published discerning the differences between the different kinds of vaccines. Strain et al. in their study identified a significant difference between those who received the Moderna vaccine compared those who received the AstraZeneca/Oxford vaccine for the key symptoms of fatigue, myalgia, and chest pain [15]. None of the other literature we studied identified any such difference and reported no association between efficacy and type of vaccines. The observational nature of these findings calls for multiple randomized controlled trials to better identify a difference.

These observations can also help us reduce vaccine hesitancy and encourage policymakers in countries where healthcare resources are scarce, to employ more feasible vaccine options to protect against the development of long COVID. This is good news when you consider the costs of mRNA vaccine procurement, storage, and distribution compared to adenovirus vector vaccines [31]. Furthermore, this may encourage international initiatives such as COVAX spearheaded by Gavi, WHO, and UNICEF [32]. The COVAX initiative has not only administered over 1 billion vaccines (mostly AstraZeneca) but has also helped develop the logistical capabilities of countries and their procurement of vaccines. In middle- and low-income countries, vaccination may act as an additional layer of protection along with the proven methods of social distancing and self-isolation in curtailing COVID and the development of long COVID.

### 4.4. Approach to Patients with Long COVID

Digital technologies have played a vital role to identify the number of patients who visited a healthcare facility for one or more symptoms of long COVID. According to one of the studies, 30% of hospitalized patients visited a healthcare facility for one or more symptoms of long COVID [33]. Another study reported 19.1% of patients being unable to work, (mostly due to a COVID-related illness.) with 12% of patients hospitalized 2 weeks after the onset of illness [28]. This highlights the urgent need for proper guidelines to identify, support and manage the symptoms of long COVID.

Currently, a comprehensive multidisciplinary approach that includes physical and mental rehabilitation services is being proposed to manage patients with long COVID. Recently, long COVID clinics, an initiative by healthcare systems, have started to emerge. Their main aim is to focus on and manage patients suffering from long COVID and to help alleviate their anxiety around their symptoms. It is recommended to put great emphasis on history and clinical evaluation and stratify patients according to their symptoms. This approach will help not only to rule out any other causative factors but also in forming a comprehensive and individualized management plan. Frequent follow-ups are advised in the long COVID clinics to better manage patients and alleviate their anxiety around their symptoms.

### 4.5. Are Antivirals an Answer?

Therapeutic options remain limited as researchers try to understand the underlying disease mechanisms and identify proper therapeutic targets. As mentioned earlier, residual viral particles and viral shedding have been proposed as probable causes of long COVID. It has been postulated that early use of antivirals can prevent viral replication, the severity of the infection, and long-term complications of coronavirus disease i.e., long COVID.

Remdesivir as an RNA-dependent RNA polymerase inhibitor blocks viral replications and has significantly proven efficacy against SARS-COV-2 infection [34]. Bigel et al. in a double-blind randomized controlled trial first reported the efficacy of remdesivir. They found that the remdesivir group had a faster recovery time as compared to the placebo. This was consistent with the findings of another trial that assessed nonhospitalized participants with a high risk of COVID-19 progression. Gottlieb et al. in this study found that a 3-day course of remdesivir had an 87% lower risk of hospitalization or death than a placebo, as well as an acceptable safety profile [35]. While the preliminary data against acute infections are promising, not much has been studied about its efficacy against long-term complications of COVID-19. The accessibility and dosing of Remdesivir also make it a difficult drug of choice to combat COVID-19.

Researchers are currently interested in the efficacy of three new oral antiviral drugs, Molnupiravir, Fluvoxamine, and PAXLOVID™, against COVID-19. Wen et al. conducted a meta-analysis to investigate their efficacy in improving mortality, hospitalization rates, and adverse events among COVID-19 patients. He reported that all three oral antivirals demonstrated no serious adverse effects and helped in preventing hospitalizations and long-term complications.

PAXLOVID™ is an investigational SARS-CoV-2 protease inhibitor antiviral therapy, specifically designed to be administered orally as soon as the symptoms of COVID-19 appear [36]. It has two components, Nirmatrelvir (Protease inhibitor) and Ritonavir (enhancer). Nirmatrelvir is a SARS-COV 2 protease inhibitor, an enzyme needed for the replication of coronavirus. Coadministration with a low dose of ritonavir helps slow the metabolism of Nirmatrelvir and increases its levels in the body. Hammond et al. in a phase 2/3 double-blind clinical trial reported an 89% lower risk of progression to severe COVID infection after treatment with PAXLOVID as compared to placebo [37]. They also reported no serious adverse effects, leading to its approval from the FDA in December 2021. Recently Najjar-Debbiny conducted a wide-scale retrospective analysis in Israel to assess its efficacy in a non-controlled setting and reported similar findings [38]. Interestingly, the study also found that PAXLOVID™ is more effective in high-risk patients; older, immunosuppressed, and patients with underlying neurological or cardiovascular disease. While PAXLOVID was associated with positive outcomes, its ritonavir component makes it a difficult drug to prescribe due to many complex drug–drug interactions. They concluded that while PAXLOVID shows great promise, the COVID-19 vaccine remains the most effective way to prevent disease progression and death among COVID-19 patients. Gupta et al. in a recent preprint reported a relapse of symptoms in a 71-year-old vaccinated and boosted male whose symptoms responded to PAXLOVID™ [39]. The viral load fluctuated with symptoms with two distinctive peaks on days 1 and 9. This poses the question if PAXLOVID™ should be used as a bridge to transiently suppress viral replication till natural immunity takes over. Further randomized controlled trials are warranted to better understand the dosing and efficacy of PAXLOVID.

Molnupiravir, another oral RNA-dependent RNA polymerase inhibitor, has recently garnered interest due to its ability to reduce viral load and clear viral particles [40]. Li et al. studied Molnupiravir in addition to PAXLOVID and their efficacy against the Omicron virus. Their study has demonstrated that molnupiravir and nirmatrelvir potently inhibited the infection of the SARS-CoV-2 Omicron variant [41]. The combination of molnupiravir and nirmatrelvir also exerted synergistic antiviral activity, indicating their efficacy against different variants.

## 5. Conclusions

In conclusion, our review summarizes the existing literature regarding the pathophysiology of long COVID, the efficacy of vaccination on long COVID, as well as current management strategies. Most of the results found vaccinated patients to report improvement in long COVID symptoms as compared to unvaccinated patients; however, the strength of the current evidence is limited, and further research needs to be carried out to identify the impact of vaccination on long COVID symptomatology. Moreover, further trials looking at individual vaccine inhibition effects on long COVID should be conducted. Studies testing the comparative efficacy of different vaccines also showed no significant difference, thereby concluding that any of the acceptable vaccines can be used. This is important for the developing world especially in terms of cost-effectiveness, where vaccine disparity is still an issue and a major limiting factor in mass vaccination. More studies testing the comparative efficacy of different vaccines should be conducted to further decrease this gap of vaccine disparity. Furthermore, physical and mental rehabilitation services and antiviral therapies show great promise in curtailing long COVID. Antivirals specifically may inhibit viral replication, reduce the severity of the infection, and may help prevent long COVID. These findings should be confirmed by conducting multiple clinical trials as they can revolutionize the therapeutic management of COVID-19.

## Figures and Tables

**Table 1 life-12-01066-t001:** Effects of vaccinations on pre-existing COVID-19 symptoms.

Author	Total Participants	Vaccine Given	Unvaccinated (*n*)	Vaccinated (*n*)	Single Dose (*n*)	Double Dose (*n*)	Participants Who Developed Long COVID (*n*)	Differences between Vaccines Reported	Vaccination Effects on Long COVID	Symptoms Effected
Arjun et al. [6]	487	Covaxin	119	368	81	287	142	N/A	More likely to develop long COVID symptoms (aOR = 2.32)	Fatigue, Fever, Chills, ENT, Respiratory
Arnold et al. [7]	66	Pfizer, AstraZeneca	22	44	N/A	N/A	66	no difference	50% less likely to develop long COVID symptoms	Respiratory, ENT, MSK, CVS, GI, Neurological Fatigue
Ayoubkhani et al. [8]	28,356	Pfizer, Moderna, Adenoviral vector vaccine	0	28,356	N/A	N/A	6729	no difference	Less likely to develop all long COVID symptoms (OR = 0.91)	Respiratory, ENT, MSK, CVS, GI, Neurological, Psychiatric, Fever, Fatigue
Gaber et al. [9]	77	Pfizer, Moderna	10	67	-	-	-	no difference	Less likely to develop 1 or more long COVID symptoms	Fatigue, Anxiety, Shortness of Breath
Kuodi et al. [10]	951	Pfizer, Moderna	317	634	340	294	337	-	36–73% less likely to develop long COVID symptoms	Fatigue, headache, MSK, DErmatological, insomnia, LOC
Nehme et al. [11]	1596	Pfizer, Moderna	825	771	424	347		no difference	Less likely to develop any one symptom (aOR 0.60)	Fatigue, Neurological, ENT, Respiratory
Peghin et al. [12]	479	Pfizer, Moderna, Astrazeneca Johnson & Johnson	347	132	N/A	N/A	201	no difference	More likely to develop ocular symptoms, less likely to develop hair loss	Hair Loss, Ocular symptoms
Scherlinger et al. [13]	567	Pfizer, moderna, Adenoviral vector vaccine	170	397	255	142	380	no difference	22% of patients felt an improvement in long COVID Symptoms	Respiratory, GI, Neurological, CVS, Psychiatric, MSK, ENT, Pruritis, Bruising, Fever, Chills
Senjam et al. [14]	773	N/A	407	366	173	193	0	N/A	Less likely to develop long COVID symptoms	ENT, Neurological, MSK, Dermatological, Fatigue, Respiratory
Strain et al. [15]	812	Astrazeneca, Pfizer, Moderna	0	812	698	114	812	MRNA superior	57% of participants reported an improvement in symptoms	Respiratory, ENT, MSK, CVS, GI, Neurological, ANS dysfunction, Rash, Fever, COVID toes
Tran et al. [16]	910	Pfizer, Moderna, Astrazeneca, Johnson & Johnson	455	455	N/A	N/A	not specified	no difference	More likely to experience remission of all long COVID symptoms (HR = 1.97)	Respiratory, ENT, MSK, CVS, GI, Neurological, Dermatological, ANS dysfunction, Psychiatric, Fatigue
Wisnivesky et al. [17]	453	Pfizer, Moderna, Jansen	129	324	N/A	N/A	not specified	no difference	No association	
	*n* = 35,527		*n* = 2801	*n* = 32,726			*n* = 8667			

N/A: not applicable.

**Table 2 life-12-01066-t002:** Effects of vaccination on long COVID in patients who were vaccinated before getting infected with COVID-19.

Author	Total Participants	Vaccinated (*n*)	Vaccine Used	Single Dose (*n*)	Double Dose (*n*)	Participants Who Developed COVID-19 (*n*)	Participants Who Developed Long COVID-19	Differences between Vaccines	Vaccination Effects on Long COVID	Symptoms Effected
Antonelli et al. [18]	4740	2370	(Pfizer, Astrazeneca Moderna)	-	2370	not specified	50% less than those with one dose or unvaccinated	None	Half as likely to have symptoms > 28 days	Respiratory, ENT, MSK, CVS, GI, Neurological, Psychiatric, Fever, Fatigue
Taquet et al. [19]	18,958	9479	65.1% were vaccinated with ‘Pfizer/BioNTech’, 9.0% with Moderna, 1.6% with Janssen’, and 24.4% with unspecified subtype	N/A	N/A	not specified	N/A	no outcomes on long COVID	Less likely to exhibit 5 symptoms of long COVID	Anosmia, Fatigue, Hairloss, Myalgias, ILD
Simon et al. [20]	240,648	20,188	Pfizer, Jansen, Moderna	2392	17,796	not specified	90,319	not reported	Less likely to have any symptom of long COVID (OR 0.22)	Respiratory, ENT, MSK, CVS, GI, Neurological, Fever, Fatigue
Blumberg et al. [21]	43	15	Pfizer	2	13	not specified	not specified	N/A	Less likely to have symptoms of long COVID.	Respiratory, CVS
*n* (Total)	*n* = 264,389	*n* = 32,052								

N/A: not applicable.

## Data Availability

Not applicable.

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
