# Peer review of "COVID-19 Vaccine and Long COVID: A Scoping Review"

_life, 2022, doi:10.3390/life12071066_

Round 1

Reviewer 1 Report

This review was described the relationship between long COVID-19 symptoms and various types of vaccines. It is interested, because readers want to know the effects of each vaccine on inhibition of long COVID-19 symptoms. Authors had collected the publications regarding long COVID-19 symptoms and made Tables. My comments were following;

Tables

Authors summarized numbers of participants with vaccination, numbers of participants with (long) COVID-19 and results of differences between vaccines, extracting from publications. However, inhibition effects between types of vaccines were not presented detail in Tables. The aim of this review is to show the differences between vaccines for long COVID-19 symptoms, and readers want to know the inhibition effects for types of vaccines on long COVID-19 symptoms easily from Table. Authors should provide Tables with real data of inhibition effects between types of vaccines for readers. The numbers of participants should be presented as compact as possible, and inhibition effects on long COVID-19 symptoms for each vaccine should be added in Tables.

Also, I found some mistakes. 

The number of vaccinated reported by Arjun et al in Table 1 will be 368.

Ayoub khani will be Ayoubkhani. That means no space. Also, the numbers of participants and vaccinated reported by Ayoubkhani et al will be 28356, and the number of no vaccinated will be 0.

I could not check mistakes between all numbers in Tables and text and the results of cited publications, because I was given only 1 week for reviewing, all numbers in Tables and text should be confirmed by several authors again, because some mistakes were found by my reviewing.

References

Authors have cited several preprints. Please ask editors whether preprints are suitable for this journal.

Author Response

Thank you for your response and words of appreciation. We have tried to address your comments by editing our tables and adding columns related to the Vaccination effect on Long COVID and the symptoms affected. We have also emphasized the same by adding this very important aspect in our conclusion. We also took this opportunity and reviewed our numbers and made some corrections. Other minor changes were also made.

Reviewer 2 Report

It is review of data on COVID-19 Vaccine and Long COVID. It provides information on various aspects of vaccination and efficacy of vaccines documented earlier providing ready reference. It is retrospective study with review of previous literature and can help researchers in future. It is well written except objectives needs to be focused and hypothesis written clearly. If statistical analysis can be included of previous work it will be good.

Author Response

Thank you for your response and comments. We reviewed our objectives again and we feel that we tried to write them very clearly and precisely. We reviewed our article and made sure that all the objectives have been addressed. To better clarify our objectives we made some additions to our conclusion as well as written below. Also, the statistical analysis of other studies has been added to the review where appropriate.

Enlisted objectives:

  • To describe the pathophysiology of Long COVID along with the most common signs and symptoms.
  • To study the impact of vaccination on patients with preexisting signs and symptoms of Long COVID.
  • To decipher the impact of vaccinations before developing COVID 19 infection.
  • To find any possible differences that may exist in the rates of developing Long COVID when comparing vaccinated vs unvaccinated people.
  • To distinguish any similarities or differences between the various types of vaccines and the risk of developing Long Covid symptoms.
  • To report the management and therapeutic options available for Long COVID.

Conclusion

‘Moreover, further trials looking at individual vaccine inhibition effects on Long COVID should be conducted’
